# Simvastatin Reduces Doxorubicin-Induced Cardiotoxicity: Effects beyond Its Antioxidant Activity

**DOI:** 10.3390/ijms24087573

**Published:** 2023-04-20

**Authors:** Michela Pecoraro, Stefania Marzocco, Raffaella Belvedere, Antonello Petrella, Silvia Franceschelli, Ada Popolo

**Affiliations:** Department of Pharmacy, University of Salerno, 84084 Fisciano, SA, Italy; mipecoraro@unisa.it (M.P.); smarzocco@unisa.it (S.M.); apetrella@unisa.it (A.P.); sfranceschelli@unisa.it (S.F.)

**Keywords:** cardiotoxicity, oxidative stress, Doxorubicin, Simvastatin, Connexin 43

## Abstract

This study aimed to evaluate if Simvastatin can reduce, and/or prevent, Doxorubicin (Doxo)-induced cardiotoxicity. H9c2 cells were treated with Simvastatin (10 µM) for 4 h and then Doxo (1 µM) was added, and the effects on oxidative stress, calcium homeostasis, and apoptosis were evaluated after 20 h. Furthermore, we evaluated the effects of Simvastatin and Doxo co-treatment on Connexin 43 (Cx43) expression and localization, since this transmembrane protein forming gap junctions is widely involved in cardioprotection. Cytofluorimetric analysis showed that Simvastatin co-treatment significantly reduced Doxo-induced cytosolic and mitochondrial ROS overproduction, apoptosis, and cytochrome c release. Spectrofluorimetric analysis performed by means of Fura2 showed that Simvastatin co-treatment reduced calcium levels stored in mitochondria and restored cytosolic calcium storage. Western blot, immunofluorescence, and cytofluorimetric analyses showed that Simvastatin co-treatment significantly reduced Doxo-induced mitochondrial Cx43 over-expression and significantly increased the membrane levels of Cx43 phosphorylated on Ser368. We hypothesized that the reduced expression of mitochondrial Cx43 could justify the reduced levels of calcium stored in mitochondria and the consequent induction of apoptosis observed in Simvastatin co-treated cells. Moreover, the increased membrane levels of Cx43 phosphorylated on Ser368, which is responsible for the closed conformational state of the gap junction, let us to hypothesize that Simvastatin leads to cell-to-cell communication interruption to block the propagation of Doxo-induced harmful stimuli. Based on these results, we can conclude that Simvastatin could be a good adjuvant in Doxo anticancer therapy. Indeed, we confirmed its antioxidant and antiapoptotic activity, and, above all, we highlighted that Simvastatin interferes with expression and cellular localization of Cx43 that is widely involved in cardioprotection.

## 1. Introduction

Doxorubicin (Doxo), an antibiotic derived from *Streptomyces peucetius* var. *caesius*, belongs to the anthracyclines family, and it is widely used to treat multiple cancer types, both solid and hematological [1].

Doxo exerts its antineoplastic activity in fast-proliferating tumors through the inhibition of the DNA topoisomerase II enzyme and DNA damage [2]. However, the effectiveness of Doxo is hampered by its main side effect, the cardiotoxicity, that is clinically characterized by rhythm disturbances, blood pressure changes, decreased ejection fraction and contractile function, cardiac dilatation, and cardiomyopathy [3]. Clinical data indicate that cardiotoxic effects increase when Doxo exceeds the cumulative dosage of 400–700 mg/m^2^ [4], but recently, increased attention has been given to Doxo-induced acute cardiotoxicity [5]. Indeed, experimental data indicate that the damage induced by Doxo on cardiomyocytes is an early event, already evident after a single administration [6], and both in vitro and in vivo, our previous studies have shown that short-term administration of Doxo is able to induce cytosolic and mitochondrial reactive oxygen species (ROS) overproduction and calcium homeostasis dysregulation and apoptosis [7,8,9], which are stated as the main causes of Doxo-induced cardiomyopathy [10]. Dexrazoxane, the only FDA-approved cardioprotective agent, showed a limited clinical application, since it has been demonstrated that it interferes with Doxo efficacy and is responsible for secondary malignancies. Furthermore, given the failure of antioxidants, β-blockers, and iron chelation therapies in treating Doxo-induced cardiotoxicity, the discovery of drugs capable of interfering in the early stages of Doxo-induced damage to cardiomyocytes could be of paramount importance [11]. It has been proven that cardiac cells put defense mechanisms in place to reduce the propagation of chemotherapeutic drug-induced cardiotoxic effects, such as variations in Connexin43 (Cx43) expression and localization [8,9].

Cx43 is a Connexin family member with phosphoproteins forming gap junctions (GJs) and non-junctional hemichannels in the ventricular myocardium. GJ permit the propagation of action potential in the heart and the cell-to-cell spread of molecules critical for cell death and survival. Beyond its role in GJ formation, Cx43 is also located in the inner membrane of mitochondria, and many studies reported that Cx43 plays a pivotal role in cardioprotection given its involvement in ROS formation [11].

Thus, Cx43 could be an attractive therapeutic target to reduce and/or prevent chemotherapeutic drug-induced cardiotoxicity. In this study, we evaluated the effects of Simvastatin (Sim) in a widely used in vitro model of Doxo-induced cardiotoxicity. Sim is an inhibitor of HMG-CoA reductase, which, like all statins, shows pleiotropic biological effects beyond its cholesterol lowering activity, so they have been widely used as cardioprotective agents in view of their antioxidant and anti-inflammatory effects [12]. Moreover, it has been proven that statins promote Cx43 and phospho-Cx43 expression in cardiomyocytes [13]. So, this study aimed to evaluate if the cardioprotective activity of Sim in a cellular model of Doxo-induced cardiotoxicity also includes an effect on Cx43 expression and/or localization.

## 2. Results

### 2.1. Effect of Simvastatin on Doxorubicin-Induced Cytosolic and Mitochondrial ROS Production

To evaluate the effects of Sim on Doxo-induced cytosolic and mitochondrial ROS production, cytofluorimetric analysis was performed by means of DHCF and MitoSOX red, respectively. As reported in Figure 1A, Doxo administration significantly (*p* < 0.0001) induced cytosolic ROS production. Sim co-treatment significantly (*p* < 0.05) reduced Doxo-induced cytosolic ROS overproduction, even if in these cells cytosolic ROS levels were significantly (*p* < 0.001) higher than control cells.

Mitochondrial ROS levels in Doxo-treated cells were significantly (*p* < 0.005) higher than control cells. Sim co-treatment significantly (*p* < 0.05) reduced Doxo-induced mitochondrial ROS overproduction (Figure 1C).

### 2.2. Effect of Simvastatin on Doxorubicin-Induced Calcium Homeostasis Dysregulation

Since calcium homeostasis impairment plays a central role in Doxo-induced cardiotoxicity, we evaluated intracellular calcium concentrations by means of FURA 2-AM in a calcium-free incubation medium (containing 0.5 mM EGTA). The mitochondrial calcium depletory, carbonyl cyanide p-trifluoromethoxyphenylhydrazone (FCCP, 50 nM), was used to evaluate mitochondrial calcium content. As reported in Figure 2A, the percentage of delta increase in mitochondrial calcium levels in Doxo-treated cells was significantly (*p* < 0.05) higher than in control cells, indicating higher levels of calcium stored in the mitochondria. In Sim co-treated cells, an evident reduction, despite not being significant, in mitochondrial calcium content was observed compared to Doxo-treated cells.

Ionomycin, a calcium ionophore, was used to evaluate cytosolic calcium content. Our data showed that, in Doxo-treated cells, the percentage of delta increase in intracellular calcium levels was lower than control cells. In Sim co-treated cells, the percentage of delta increase in intracellular calcium levels was significantly (*p* < 0.05) higher than Doxo-treated cells, indicating an improvement in intracellular calcium storage (Figure 2B).

### 2.3. Effect of Simvastatin on Doxorubicin-Induced Apoptosis

It has been well-established that ROS overproduction is the leading cause of Doxo-induced apoptosis [4]. So, to analyze the cardioprotective effects of Sim co-treatment, we evaluated apoptosis by cytofluorimetric analysis of PI-stained hypodiploid nuclei and cytochrome c release in our experimental model. Our results showed that Doxo administration significantly (*p* < 0.005) induced the apoptotic response, as shown by the percentage of the hypodiploid nuclei. Sim co-treatment significantly (*p* < 0.05) reduced the pro-apoptotic effects of Doxo (Figure 3A).

Similarly, cytosolic cytochrome c levels in Doxo-treated cells were significantly (*p* < 0.005) higher than control cells, while in Sim co-treated cells, a significant (*p* < 0.05) reduction in cytochrome c release was observed compared to Doxo-treated cells (Figure 3C).

### 2.4. Effect of Simvastatin on Cx43 and pCx43 Expression and Localization

It has been reported that changes in Cx43 expression and localization are adaptive mechanisms implemented by cardiac cells to counteract chemotherapeutic agent-induced damage [9]; then, we evaluated the effects of Sim co-treatment on this protein. In our experimental model, we showed that Doxo administration induced a significant (*p* < 0.0001) increase in intracellular Cx43 levels, evaluated by means of cytofluorimetry in permeabilized conditions. The same result has been observed in Sim co-treated cells (Figure 4A).

Western blot analysis performed on mitochondrial lysates showed a significant (*p* < 0.05) increase in mCx43 expression in Doxo-treated cells. Sim co-treatment significantly (*p* < 0.05) reduced Doxo-induced mCx43 overexpression (Figure 4C).

The confocal analysis highlighted these results. In detail, the signal for Cx43, which is typically at plasma membrane and partially in cytosol, became nuclear (panels a and d for single and merged signals, respectively). On the other hand, the images obtained in presence of Sim showed no significant changes in terms of Cx43 cellular localization if compared to control cells (panels i, j, and l for Cx43, VDAC1, and merged signals in Sim-treated cells and panels a, b, and d for the same signals in the control experimental point, respectively). Interestingly, in Sim co-treated cells, Cx43 translocated from the plasma membrane to cytosol, where it appeared to co-localize with VDAC1 protein as detected in panel p (indicating the merged signals of panels m, n, and o; see white arrows) (Figure 4D).

Cx43 membrane levels were evaluated by using cytofluorimetry in non-permeabilized conditions. Our data showed a significant (*p* < 0.001) increase in Cx43 membrane levels both in Doxo-treated cell and in Sim co-treated cells (Figure 5A). It is important to note that Sim co-treatment significantly (*p* < 0.005) increased the membrane levels of Cx43 phosphorylated on Ser368 compared to Doxo-treated cells (Figure 5C).

## 3. Discussion

Doxorubicin (Doxo) is an anticancer drug belonging to the anthracycline family that is broad-spectrum and highly effective for antineoplastic agents widely used in the various cancer treatments, including breast cancer, ovarian cancer, leukemias, lymphomas, and pediatric neoplasms. Doxo acts through several mechanisms of action: intercalation of DNA, inhibition of topoisomerase II, damage to cell membranes with consequent alteration of permeability to calcium ion, and ROS production [2]. The last two action mechanisms seem to be responsible for the main side effect of drug, the cardiotoxicity, which affects nearly 30% of patients within 5 years of chemotherapy. This cumulative and dose-dependent toxicity greatly hampers the success of clinical Doxo applications. Heart failure in Doxo-treated patients may go undetected for many years, since the time interval between treatment and the onset of the chronic form can vary from 30 days to more than 10 years [4,14]. Many studies, performed to evaluate the underlying mechanisms of Doxo-induced cardiotoxicity, confirmed that they are both complex and multifactorial [5]. For these reasons, the knowledge regarding the leading cause of Doxo-induced acute cardiac injury and the search for viable therapeutic strategy that can reduce, but also prevent, this side effect is of paramount importance. Recent data suggest that statins could be promising agents to counteract Doxo-induced cardiotoxicity. Statins are reversible and competitive inhibitors of the enzyme HMG-CoA reductase, but they have shown some other effects, termed “pleiotropic”, which are ascribed to their ability to regulate the function of small GTPases of the Rho family. These effects, in fact, are mediated by the inhibition of isoprenoid intermediates, which are responsible for post-translational modifications of these small GTP-binding proteins and involve the following: (i) the decrease of oxidative stress through the reduction of ROS levels; (ii) the reduction of inflammation by preventing the dislocation of IkB from NF-kB, resulting in its inactivation; and (iii) the reduction of cardiac hypertrophy risk [15,16].

These mechanisms underlie the beneficial effects of statins on the cardiovascular system and could also counteract chemotherapy-induced cardiotoxic complications [17]. Based on these observations, this study aimed to evaluate the effects of Sim co-treatment on Doxo-induced cardiotoxicity in an in vitro model. We used the rat cardiomyoblast cell line (H9c2), since these cells are widely used to study the possible cardiotoxic effects of anticancer drugs in view of their similarities with primary cardiomyocytes [18,19].

Since oxidative stress and calcium homeostasis impairment seem to be responsible for Doxo-induced cardiotoxicity, as a first step, we evaluated the effects of Sim co-treatment on these parameters in our experimental model, designed on the basis of a previous study [20].

It is well-known that Doxo results in increased oxidative stress through multiple pathways, including redox cycling of the molecule and formation of ionic complexes with iron. The final effect is the generation of highly toxic ROS that react with any oxidizable compounds, thereby inducing damage to lipids, nucleic acids, and proteins [21,22]. Our data show that Sim co-treatment significantly reduces Doxo-induced over-production of both cytosolic and mitochondrial ROS, thus confirming the antioxidant activity of statins [23,24].

Doxo-induced cardiotoxicity is also characterized by alterations in calcium homeostasis, since it has been demonstrated that Doxo affects the expression levels of calcium regulatory proteins expressed on the plasma membrane [25,26] and on the sarcoplasmic reticulum [27,28], and it also interferes with the regulation of mitochondrial Ca^2+^ levels. Indeed, it has been reported that, following Doxo administration, there is an initial accumulation of Ca^2+^ in the mitochondria, in order to reduce its cytoplasmic concentration, thus limiting the detrimental effects exerted by high levels of this cation in the cytosol [29,30]. Our data confirm these notions, since we reported an increase of mitochondrial calcium levels in Doxo-treated cells. Furthermore, we showed that Sim co-treatment restores mitochondrial Ca^2+^ levels and induces an increase in Ca^2+^ levels stored in cytosolic stores. These results are consistent with previous studies reporting that statins can affect Ca^2+^ stores in cardiomyocytes; specifically, statins increase SERCA2a protein expression and the consequent Ca^2+^ storage in the RER, thus reducing the severity of myocardial infarction in rats [31,32]. Moreover, it has been shown that statins reduce intramitochondrial calcium and increase mitochondrial transmembrane potential, preventing PTP pore opening and cytochrome c release [33].

Increased oxidative stress and changes in Ca^2+^ homeostasis within the cell are known to be precursors of the apoptotic process [34,35]. Indeed, Doxo-induced mitochondrial Ca^2+^ overload, also observed in our experimental model, can switch from an initially physiological and defensive process into a lethal event, since it stimulates the mitochondrial permeability transition pore opening, resulting in the inner mitochondrial membrane depolarization, mitochondrial swelling, and outer membrane disruption, which eventually induce the release of cytochrome c into the cytosol and the consequent activation of the caspase pathway [36]. Our results show that Sim co-treatment significantly reduces Doxo-induced apoptosis and cytochrome c release, thus corroborating the anti-apoptotic activity of statins reported in the literature [37,38].

Previous studies report that, following Doxo treatment, cardiomyocytes put compensatory mechanisms in place that try to limit damage; among these, an important role is played by Cx43 [8]. Cx43 is a transmembrane phosphoprotein critical for the establishment of gap junctions (GJs), intercellular structures that enable communication between cells, allowing for the passage of ions, low-molecular-weight metabolites, and signal molecules. Specifically, in the heart, GJs facilitate the conduction of the action potential between cardiomyocytes and synchronize atrioventricular contraction, ensuring the proper maintenance of heart rhythm [39]. In addition, some other functions have been reported for Cx43 that go beyond its role in intercellular communication. Some of these functions have been related to the presence of Cx43 in the inner mitochondrial membrane (mCx43). The role of mCx43 has not been fully elucidated; however, strong evidence indicates that it modulates mitochondrial respiration, ROS production, ATP synthesis, and K^+^ influx via mitochondrial ATP-dependent K^+^ channels [40].

Recent studies suggest that some of the protective effects of statins might be related to modulation of connexins in cardiomyocytes [41]. Indeed, evidence shows that statins could increase myocardial Cx43 expression, which is downregulated in certain pathological conditions, such as hypertriglyceridemia, acute myocardial infarction, viral myocarditis, or diabetes [13,42,43]; suppress GJ remodeling and correct the disordered distribution of connexins typical of heart failure [13,44,45]; and modulate the phosphorylation of the connexins, which is related to their internalization and degradation [13].

In view of the pivotal role played by Cx43 in cardioprotection and in the adaptive response to Doxo-induced damage, we evaluated the effects of Sim co-treatment on this protein expression and localization in our experimental model.

Our data showed that Doxo administration induces a significant increase in intracellular Cx43 levels, and Sim did not interfere with this effect. Immunofluorescence analysis confirmed a change in the intracellular Cx43 localization, probably due to an increase in the synthesis and/or in the trafficking of this protein. Furthermore, Western blotting analysis showed that Doxo induced a significant increase in mCx43 expression, in agreement with previous studies demonstrating that mCx43 overexpression is a defense mechanism aimed to reduce Doxo-induced damage in cardiac cells [8,9,46]. mCx43 is involved in calcium homeostasis, since it appears to increase the uptake of calcium ions into the mitochondrion, storing them internally and thereby delaying the increase in cytosolic calcium levels that can trigger the apoptotic pathway [47]. The significant reduction of mCx43 expression observed in Sim co-treated cells in our experimental model could account for the lower levels of calcium stored in mitochondria and for the reduced induction of apoptosis observed in these cells. It is reasonable that Sim reduces the free calcium levels in the cells, so they do not necessitate of the so-called “sponge effect” carried out by mitochondria [48].

As previously mentioned, Cx43 is a phosphoprotein that is predominantly phosphorylated in the control state. It has been reported that, at least 17 serine sites and 2 tyrosine sites that could be phosphorylated via several kinases are located at the C-terminus of Cx43. The level of kinase activation and related Cx43 expression and phosphorylation affects GJ conductance. PKC activation, in particular, which phosphorylates Ser368, has been shown to decrease GJ communication [49,50].

In our experimental model, we showed that, while Sim co-treatment did not affect Cx43 membrane levels, it significantly increased the levels of Cx43 phosphorylated on Ser368. We hypothesize that this effect plays a pivotal role in cardioprotection and could account for the observed reduction of apoptosis spreading observed in these cells. Indeed, this hypothesis is in agreement with previous study demonstrating that the disruption of cell–cell communication through GJs is a defense mechanism implemented by cells, known as the “Good Samaritan” mechanism in that it blocks the propagation of harmful stimuli to healthy cells [51,52].

Based on these results, we can conclude that Sim could be a good adjuvant in antitumor therapy with Doxo. Indeed, we confirmed that Sim can prevent cardiotoxicity by virtue of its well-known antioxidant activity, but here we also highlighted its effects on mitochondrial and membrane levels of Cx43, a protein mainly involved in cardioprotection and in the first phase of adaptive response to Doxo-induced damage.

The main limitation of this study is that these detections were made in an in vitro model. Further in vivo studies are needed to better corroborate our conclusions.

## 4. Materials and Methods

### 4.1. Reagents

The monoclonal antibodies used were anti-Cx43 (C6219; Sigma, Milan, Italy), anti-pCx43 (sc-17219, Santa Cruz, Dallas, TX, USA), anti-TOM20 (C-11415, Santa Cruz), anti-VDAC1 (sc-8829, Santa Cruz), and anti-cytochrome c (sc-13156, Santa Cruz). Secondary antibodies (anti-rabbit, A120-101P and anti-mouse, A90-137P) were purchased from Bethyl Laboratories (Montgomery, TX, USA). Texas red- and fluorescein-conjugated secondary antibodies (T6390 and F-2761) were bought by Thermo Fisher Scientific (Waltham, MA, USA). MitoSOX Red Mitochondrial Superoxide Indicator (M36008) was bought from Invitrogen, ThermoFisher. FCCP (C2920), H2DCF-DA (D6883), FURA 2AM (F0888), Ionomicin (I3909), and propidium iodide (4170) were purchased from Sigma-Aldrich (St. Louis, MO, USA). ECL (RPN2209) was bought from Sigma (Milan, Italy). AlexaFluor anti-mouse 488 (A32723), anti-rabbit 555 (A32732), and DAPI (D1306) were bought by Molecular Probes (Eugene, OR, USA).

### 4.2. Materials

Doxorubicin (S-5040420001) and Simvastatin (S6196) (both from Sigma—Italy) were used. Embryonic rat heart cardiomyocyte-derived cell line H9c2 (from the American Tissue Culture Collection; Manassas, VA, USA) was grown adherent to Petri dishes with Dulbecco’s modified Eagle’s medium (DMEM), which was supplemented with 10% fetal bovine serum (FBS), 25 u/mL penicillin, and 25 u/mL streptomycin at a humidified air with 5% CO_2_ and 37 °C in an incubator to enable the cells to attach and reach the exponential phase of growth.

### 4.3. Cell Treatment

H9c2 cells were plated at a density required for each of the different experimental analysis and, after 24 h adhesion, were treated with Sim (10 µM) for 4 h and then co-exposed to Sim and Doxo (1 µM) for 20 h.

### 4.4. Measurement of Intracellular Reactive Oxygen Species (ROS)

Cytosolic ROS production was evaluated in H9c2 cells (4.5 × 10^5^ cells/well) plated into 6-well plates and treated as previously described. After treatment, cells were collected, washed twice with phosphate buffer saline (PBS), and then incubated in PBS containing the probe 2′,7′-dichlorofluorescin diacetate (H_2_DCF-DA; 10 μM) at 37 °C for 45 min. H_2_DCF, derived from H_2_DCF-DA cleavage by intracellular esterases, is rapidly oxidized to the highly fluorescent DCF in the presence of intracellular ROS. Cell fluorescence was then evaluated by means of a fluorescence-activated cell sorting (FACSscan; Becton–Dickinson) and analyzed with Cell Quest software (version number 5.2.1) [53].

### 4.5. Measurement of Mitochondrial Superoxide Formation

Mitochondrial superoxide formation was evaluated in H9c2 cells (4.5 × 10^5^ cells/well), plated in 6-well plates, and treated as previously described. After treatment, the fluorogenic dye MitoSOX Red (2.5 μM) was added for 15 min at 37 °C, and then cells were washed gently with PBS and collected for fluorescence evaluation by means of flow cytofluorometry. As previously described [53], once targeted to mitochondria of living cells, MitoSOX Red is readily oxidized by superoxide, but not by other ROS-generating systems, and exhibits red fluorescence. Cell fluorescence was evaluated by means of FACS scan and analyzed by Cell Quest software.

### 4.6. Measurement of Intracellular Calcium Signalling

Intracellular calcium concentrations were measured by means of the fluorescent indicator dye Fura 2-AM, the membrane-permeant acetoxymethyl ester form of Fura 2, in H9c2 cells (3 × 10^4^ cells/plate), plated in 86 mm tissue culture plates, and treated as previously described. After treatment, cells were washed in phosphate-buffered saline (PBS) and re-suspended in 1 mL of Hank’s balanced salt solution (HBSS) containing 5 μM Fura 2-AM for 45 min. Thereafter, cells were washed with HBSS to remove excess Fura 2-AM and incubated for 15 min in calcium-free buffer (HBSS containing 0.5 mM EGTA) to allow for the hydrolysis of Fura 2-AM into its active-dye form, Fura 2. H9c2 cells then were transferred to the spectrofluorometer (Perkin-Elmer LS-55). Carbonyl cyanide p-trifluoromethoxy-pyhenylhydrazone (FCCP, 50 nM final concentration) or Ionomycin (1 μM final concentration) was added into the cuvette in calcium-free buffer in order to evaluate mitochondrial calcium levels and cytosolic calcium levels, respectively. As previously reported [52,54], the ratio of fluorescence intensity of 340/380 nm (F340/F380) is strictly related to intracellular free calcium, so the excitation wavelength was alternated between 340 and 380 nm, and emission fluorescence was recorded at 515 nm. Data were expressed as the percentage of delta (% Δ) increase of the fluorescence ratio (F340/F380 nm) induced by FCCP (50 nM) or Ionomycin (1 μM)—basal fluorescence ratio (F340/F380 nm)/basal fluorescence ratio (F340/F380 nm).

### 4.7. Analysis of Apoptosis

Apoptosis analysis was performed by means of Propidium Iodide (PI), a fluorochrome capable of binding cellular DNA content in H9c2 cells (4.5 × 10^5^ cells/well) plated in a 6-well plate and treated as previously described. After treatment, cells were washed twice with PBS and incubated at 4 °C for 30 min in the dark in 500 μL of a solution containing 0.1% Triton X-100, 0.1% sodium citrate, and 50 μg/mL PI. The PI-stained cells were analyzed by means of FACS using CellQuest software [55]. Data are expressed as the percentage of cells in the hypodiploid region.

### 4.8. Flow Cytometry Analysis

In order to assess membrane levels of Connexin 43 (Cx43) and Connexin 43 phosphorylated on Ser368 (pCx43), H9c2 cells (4.5 × 10^5^ cells/well) were plated in a 6-well plate and treated as previously described. After treatment, cells were collected with scraper and treated with fixing buffer (containing 4% formaldehyde, 2% FBS, and 0.1% NaN_3_ in PBS) for 20 min. Subsequently, cells were incubated with Cx43 (1:250) or pCx43 (1:250) antibody and the appropriate secondary antibody (anti-rabbit or anti-mouse FITC antibody, both 1:2000, ThermoFisher, Waltham, MA, USA) for 1 h at 4 °C. To assess the intracellular levels of Cx43, pCx43, and cytochrome c, cells were permeabilized with fix perm buffer (fixing buffer containing 0.1% Triton X) for 30 min and then anti-cytochrome c (1:250)*,* anti-cx43 (1:250), or anti pCx43 (1:250) antibody and the appropriate secondary antibody (anti-rabbit or anti-mouse FITC antibody) were added. (1:2000) for 1 h at 4 °C. Cells collected were evaluated by fluorescence-activated cell sorting (FACSscan; Becton-Dickinson, Milan, Italy), and data obtained were analyzed by means of Cell Quest software [52]. Results are shown as the percentage of positive cells.

### 4.9. Mitochondrial Protein Extraction and Western Blot Analysis for Mitochondrial Cx43

H9c2 (1 × 10^6^ cells/well) were seeded into Petri plates and treated as described. Mitochondrial protein extraction was carried out from cells in lysis buffer A (1 mM EGTA, 1 mM EDTA, 10 mM KCl, 250 mM sucrose, 1.5 mM MgCl_2_, 50 mM NaF, 0.2 mM Na_3_VO_4_, 1 mM PMSF, 1 mM DTT, protease inhibitors, digitonin 0.025%, 20 mM K^+^ Hepes pH 7.5). After centrifugation at 16,000× *g* for 2 min at 4 °C, the supernatant was discarded, and the pellet was resuspended in lysis buffer B (0.1% Triton X, 0.5% NaDeOH, 1% SDS, 150 mM NaCl, and 50 mM Tris HCl at pH 7.4) to obtain mitochondrial protein. Protein concentrations were estimated by the Bio-Rad protein assay (BIO-RAD, Milan, Italy) using bovine serum albumin as the standard. Equal amounts of protein (50 μg) were loaded into an acrylamide gel and separated by SDS-PAGE under denaturing conditions. Blots were incubated with primary antibody anti-Cx43 (1:8000) or anti-TOM20 (1:250; used as loading control) overnight. Blots were then washed in PBS/0.1% Tween and incubated with the appropriate secondary antibody, anti-rabbit or anti-mouse (each diluted 1:4000), for 1 h at room temperature. Immunoreactive protein bands were visualized using enhanced chemiluminescence reagents (ECL) in LAS 4000 (GE Healthcare, Milan, Italy). The images were analyzed for densitometry using ImageJ Software (version number 1.44) [52].

### 4.10. Immunofluorescen Analysis with Confocal Microscopy

H9c2 cells (2 × 10^4^ cells/well) were plated in a 6-well plate and treated as previously described. After treatment, cells were fixed in 4% *v*/*v* p-formaldehyde, permeabilized with 0.4% *v*/*v* Triton X-100, and blocked with 20% *v*/*v* goat serum as previously reported [56]. Next, cells were incubated overnight at 4 °C with mouse monoclonal antibody against connexin 43 (1:250) and rabbit polyclonal antibody against VDAC1 (1:100). After that, AlexaFluor anti-mouse 488 and anti-rabbit 555 (1:500) were used for 2 h at room temperature (RT) in the dark. Nuclei were detected thanks to Dapi (1:1000). The coverslips were vertically scanned from the bottom by using a 63× (1.40 NA) Plan-Apochromat oil-immersion objective. Images, shown as a single stack, and scale bars were generated with Zeiss ZEN Confocal Software (version number Zen 3.6; Carl Zeiss MicroImaging GmbH, Jena, Germany).

### 4.11. Statistical Analysis

Data are reported as mean ± S.E.M. of at least three independent experiments, each performed in duplicate. Statistical analysis was performed by one-way analysis of variance (ANOVA), followed by Bonferroni post-test, using GraphPad Prism7 (GraphPad Software Inc., San Diego, CA, USA). A *p*-value lower than 0.05 was considered statistically significant.

## Figures and Tables

**Figure 1 ijms-24-07573-f001:**
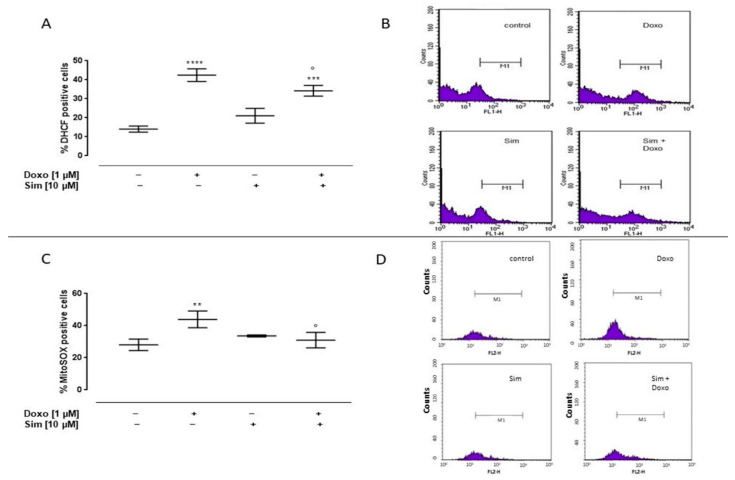
Effect of Simvastatin on Doxorubicin-induced cytosolic and mitochondrial ROS production. H9c2 cells were pre-treated with Sim (10 µM) for 4 h and then co-exposed to Sim and Doxo (1 µM) for 20 h. The fluorescent probe 2′,7′-dichlorofluorescein diacetate (H_2_DCF-DA) was used to evaluate cytosolic ROS generation by flow cytometry analysis. Results are reported as mean ± S.E.M. of DCF-positive cells’ percentage of at least three independent experiments, each performed in duplicate (**A**). The fluorescent probe MitoSOX was used to evaluate mitochondrial superoxide production by flow cytometry analysis. Results are reported as mean ± S.E.M. of MitoSOX-positive cells’ percentage of at least three independent experiments, each performed in duplicate (**C**). Representative histograms for the flow cytometry analysis are reported in Panels (**B**,**D**), respectively. Data were analyzed by variance test analysis, and multiple comparisons were made by Bonferroni’s test. ** *p* < 0.005, *** *p* < 0.001, and **** *p* < 0.0001 vs. untreated cells; ° *p* < 0.05 vs. Doxo-treated cells.

**Figure 2 ijms-24-07573-f002:**
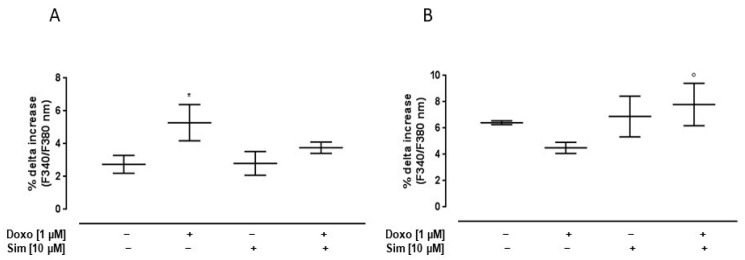
Effect of Simvastatin on Doxorubicin-induced calcium homeostasis dysregulation. H9c2 cells were pre-treated with Sim (10 µM) for 4 h and then co-exposed to Sim and Doxo (1 µM) for 20 h. FCCP (50 nM) in calcium-free medium was used to evaluate mitochondrial calcium levels (**A**). Ionomycin (1 μM) in calcium-free medium was used to evaluate intracellular calcium levels (**B**). Results are reported as mean ± S.E.M. of percentage of delta (% Δ) increase in FURA2 ratio fluorescence (340/380 nm) from at least three independent experiments, each performed in duplicate. Data were analyzed by variance test analysis, and multiple comparisons were made by Bonferroni’s test. * *p* < 0.05 vs. untreated cells; ° *p* < 0.05 vs. Doxo-treated cells.

**Figure 3 ijms-24-07573-f003:**
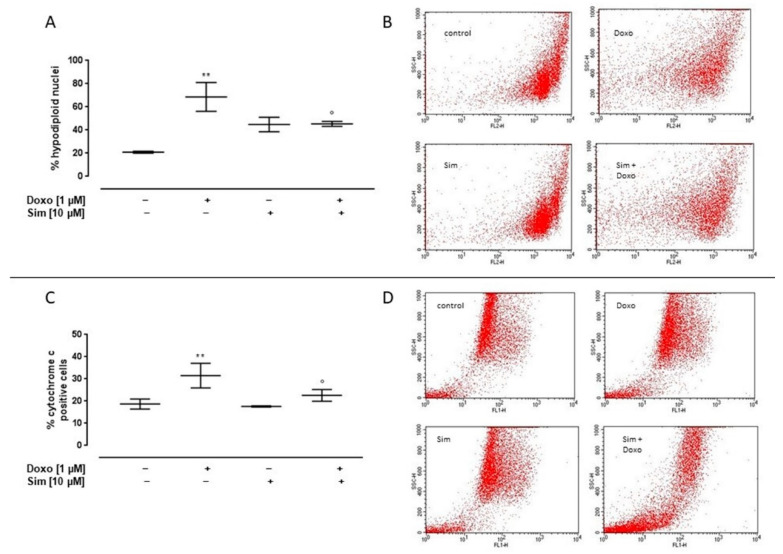
Effect of Simvastatin on Doxorubicin-induced apoptosis. H9c2 cells were pre-treated with Sim (10 µM) for 4 h and then co-exposed to Sim and Doxo (1 µM) for 20 h. Cells were stained by propidium iodide, and fluorescence of individual nuclei was measured by flow cytometry. Results are reported as mean ± S.E.M. of percentage of hypodiploid nuclei from at least three independent experiments, each performed in duplicate (**A**). Flow cytometry analysis was used to evaluate cytosolic cytochrome c level (**C**). Results are reported as mean ± S.E.M. of cytochrome c-positive cells’ percentage from at least three independent experiments, each performed in duplicate. Representative histograms for the flow cytometry analysis are reported in panels (**B**,**D**), respectively. Data were analyzed by variance test analysis, and multiple comparisons were made by Bonferroni’s test. ** *p* < 0.005 vs. untreated cells; ° *p* < 0.05 vs. Doxo-treated cells.

**Figure 4 ijms-24-07573-f004:**
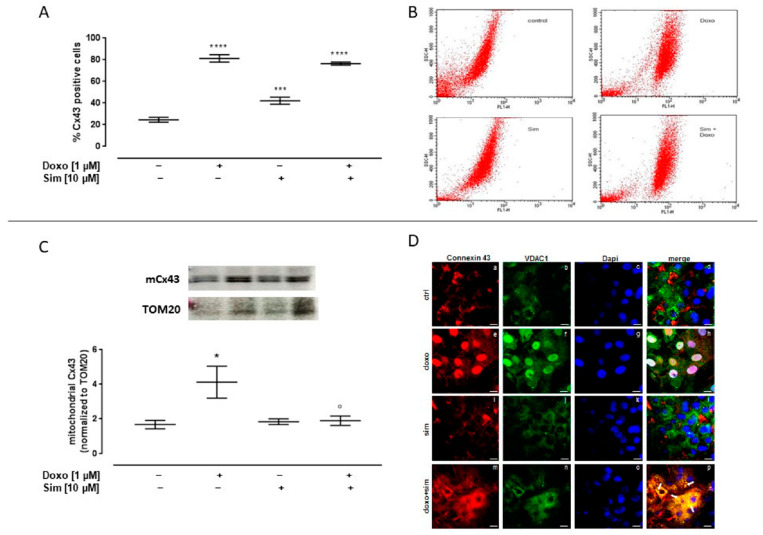
Effect of Simvastatin on intracellular Connexin43 expression. H9c2 cells were pre-treated with Sim (10 µM) for 4 h and then co-exposed to Sim and Doxo (1 µM) for 20 h. Intracellular Cx43 level was detected by flow cytometry analysis. Results are reported as mean ± S.E.M. of Cx43-positive cells’ percentage from at least three independent experiments, each performed in duplicate (**A**). Representative histograms for the flow cytometry analysis (**B**). Mitochondrial Cx43 expression was detected by Western blot analysis, and TOM20 expression was used as the loading control. Results are expressed as mean ± S.E.M from at least three independent experiments, each performed in duplicate (panel (**C**). Cells were stained with Cx43 (red) (panels **a**,**e**,**i**,**m**), VDAC1 (green) (panels **b**,**f**,**j**,**n**), and nucleus with DAPI (blue) (panels **c**,**g**,**k**,**o**) and were determined by immunofluorescence analysis (merge in panels **d**,**h**,**l**,**p**). Scale bar, 50 μm. A representative of three experiments is shown (**D**). Data were analyzed by variance test analysis, and multiple comparisons were made by Bonferroni’s test. * *p* < 0.05, *** *p* < 0.001, and **** *p* < 0.0001 vs. untreated cells; ° *p* < 0.05 vs. Doxo-treated cells.

**Figure 5 ijms-24-07573-f005:**
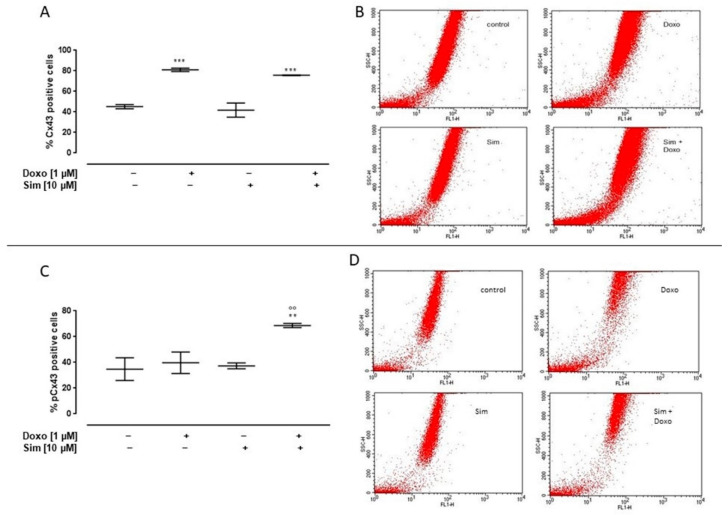
Effect of Simvastatin on membrane levels of Connexin43 and Connexin43 phosphorylated on Ser368. H9c2 cells were pre-treated with Sim (10 µM) for 4 h and then co-exposed to Sim and Doxo (1 µM) for 20 h. Cx43 (**A**) and pCx43 (**C**) levels were detected by flow cytometry analysis. Results are reported as mean ± S.E.M. of Cx43- and pCx43-positive cells’ percentage from at least three independent experiments, each performed in duplicate. Representative histograms for the flow cytometry analysis are reported in Panels (**B**,**D**), respectively. Data were analyzed by variance test analysis, and multiple comparisons were made by Bonferroni’s test. ** *p* < 0.005 and *** *p* < 0.001 vs. untreated cells; °° *p* < 0.005 vs. Doxo-treated cells.

## Data Availability

The data presented in this study are available on request from the corresponding author.

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
