# Peer review of "Simvastatin Reduces Doxorubicin-Induced Cardiotoxicity: Effects beyond Its Antioxidant Activity"

_ijms, 2023, doi:10.3390/ijms24087573_

Round 1

Reviewer 1 Report

Dear authors, 

After analysis of your manuscript, I consider that some issues should be addressed:

Comment 1. The main observation concerns the in vitro model of Doxo-induced cardiotoxicity.

a.     What was the rationale for the dose used of Doxo (1µM) and duration (20 hours) in the experiment? Another study (Pecoraro et al. 2015 Cardiovascular Toxicil) suggests that 5 µM of Doxo during 24 hours is the optimizing treatment for Doxo-induced cardiotoxicity in the H9c2 cell line!  The choice of Doxo dose and duration of treatment is based on early validation. How to prove 1 µM DOX induce cell injury?

b.     Moreover, on page 2 “Clinical data indicate that cardiotoxic effect increase when Doxo exceeds the cumulative dosage of 400-700 mg/m2”. Does the dose used in the H9c2 line have clinical significance?

c.     What was the rationale for the Simvastatin (10 µM) dose and duration (4 hours)?

d.     Cell viability and cytotoxic test are missing to prove the DOX and Sim effect in the cell line.

e.     A clear description of the experimental scheme in vitro model is missing.  It is not reported the total number of cells used in cell treatment (section 4.2 cell treatment).

In my opinion, in the absence of these data, it is not possible to formulate reliable conclusions.

Comment 2. The section Material and Methods should be more descriptive and informative. Please rewrite. A lot of information is missing.

a. In this section, the information related to reagents, drugs, and chemicals used in the experiment should be mentioned, namely information related to purchase and purity, and catalog number.

b. Moreover, in this section, the information related to antibodies should be introduced (catalog number, dilution,…)

c.     Finally, I would like to see the whole membranes of each western blotting realized. Please introduce all whole membranes in supplementary data.

Comment 3. The abstract needs to be improved. Information regarding in vitro model of DOXO-induced cardiotoxicity is missing. Information regarding cell cultures is missing. Several important pieces of information are missing. Needs to rewrite concisely.

Comment 4. A careful check of the bibliography should be done.

Author Response

All answers are reported in the attached file 

Reviewer 2 Report

Dear Authors

The experiment is well designed and conducted, however, there is a fundamental problem, and it is that the novelty of the work is low, because in this field, two similar articles have been published as below. Therefore, this manuscript can be published if the authors provide sufficient reasons for their work being different from these researchers:

1-Abd Elbaky, N. A., Ali, A. A., & Ahmed, R. A. (2010). Cardioprotective effect of simvastatin on doxorubicin induced oxidative cardiotoxicity in rats. Journal of Basic and Applied Sciences6(1), 29-38.

2-Riad, A., Bien, S., Westermann, D., Becher, P. M., Loya, K., Landmesser, U., ... & Tschöpe, C. (2009). Pretreatment with statin attenuates the cardiotoxicity of Doxorubicin in mice. Cancer research69(2), 695-699.

  .....

Author Response

All answers to referee are reported in the attached file

Reviewer 3 Report

The paper titled: “Simvastatin reduces Doxorubicin-induced cardiotoxicity: effects beyond its antioxidant activity” submitted by the authors Pecoraro et al., studied  the cardioprotective effects of Simvastatin in a cellular model of Doxo-induced cardiotoxicity and its role on Cx43 expression and/or localization. The paper investigated important topic in its field of research.

There are some things need to be addressed before the publishing of this paper:

1.       The abstract: it need to be joined into one short paragraph summarizing the objective, methodology, results and importance.

2.       In the introduction:

-          The novelty of the work had been clearly highlighted.

3.       The results 

-          Figure 1 and figure 3 are blurry and need to be sharpened and enhanced

-          Figures 4 and 5 flowcytometry photos are blurry and need to be reproduced with high quality.

4.       The discussion: There are old references need to be removed. Also in the introduction part  

All old references than 2010 should be removed and focus on the last 5 years.

5.       The materials and methods part   

-           In the paragraph : 4.3 Measurement of intracellular Reactive Oxygen Species (ROS), please add the reference of the method.

-          In the paragraph 4.5 Measurement of intracellular calcium signaling, please add the reference of the method.

-          In the paragraph 4.6 Analysis of apoptosis, please add the reference of the method.

-          In the paragraph 4.7 Flow Cytometry Analysis, please add the reference of the method.

-          In the paragraph 4.8 Mitochondrial protein extraction and Western blot analysis for mitochondrial Cx43, please add the reference of the method.

6.       The conclusion is missing. You need to show the major results found in the study and the prospect of the work in the conclusion.  

7.       References: Old references (such as those in 2001) need to be removed, please try to focus on the last 5 years.

I give you major revision.

Author Response

(The authors gave the same response as above.)

Round 2

Reviewer 2 Report

Dear Authors

Your answer to the basic problem of the low novelty of the work has not yet convinced me because the work of your article is not very different from the similar article published in this field that was done even under better conditions, i.e. in vivo conditions (Attached article). You reasoned that you also investigated the effect of SIM on the expression of the CX43 gene and its localization. This paper (Riad et al., 2009) also examined the effect of Statin + DOx on expression of similar genes including Bcl-2, BAX and TNF-alpha. 

Author Response

It is well known that the underlying mechanisms of Doxo-induced cardiotoxicity are both complex and multifactorial. Despite oxidative stress has been indicated as one of the main culprits, clinical practice has shown that the use of antioxidants (e.g. dexrazoxane) is not enough to limit the doxorubicin-induced heart damage.

For this reason, the research of all the mechanisms involved in cardiotoxicity is of paramount importance in order to define therapeutic strategies capable of preventing and/or limiting cardiac damage.

In this context the paper cited by the referee (Riad et al., Cancer Res 2009) is a milestone, as demonstrated by his numerous citations (142). Among all, we count:

  • Kuseu et al., Drug and Chemical Toxicology46 (2), pp.400-411 Mar 4 2023
  • Oh et al., Journal of Molecular and Cellular Cardiology 138, pp.244-25 Jan 2020
  • Alimoradian et al., Physiology and Pharmacology 22 (1), pp.63-72 Win 2018
  • Henninger etv al., Pharmacological Research 91, pp.47-56 Jan 2015 | 
  • Ramanjaneyulu et al., Journal of Physiology and Biochemistry 69 (3), pp.513-525 Sep 2013  
  • Kim et al., Toxicology Mechanisms and Methods 22 (6), pp.488-495 Jul 2012

and numerous reviews, highlighting the continued research in this field even after the publication of the paper mentioned.

Moreover, we well known this paper, since we also cited it in our previous study (Pecoraro et al., 2016. Toxicology and Applied Pharmacology 293: 44-52).

Our present paper also fits in this context.

Compared to the work of Riad there are many differences.

In their paper, Riad and co-workers evaluated the effects of Fluvastatin in an in vivo model of Doxorubicin-induced cardiotoxicity. They studied the effects of Statin pre-treatment on left ventricular function and also evaluated lipid peroxidation activity, SOD (1-2-3), Bax, Bcl2 and TNF-α expression.

In our study, we evaluated the effects of Simvastatin co-treatment in a cellular model of Doxorubicin-induced cardiotoxicity. We studied the effects of Simvastatin on Doxorubicin-induced oxidative stress assessing cytosolic and mitochondrial ROS levels, we also evaluated intracellular calcium levels (since Doxorubicin induces calcium homeostasis dysregulation), apoptosis and cytosolic cytochrome c levels. These analyses have led us to confirm the antioxidant and antiapoptotic activities of Statins (as stated in the paper). But here we also showed that Simvastatin reduces Doxorubicin-induced calcium homeostasis dysregulation.

Moreover, in our study we firstly focused our attention on the effects of Simvastatin co-treatment on Connexin43 (Cx43) expression and/or localization.

Cx43 is a member of connexin family, phosphoprotein involved in Gap Junction formation, intercellular structures that facilitate the conduction of the action potential between cardiomyocytes and synchronize atrioventricular contraction, ensuring the proper maintenance of heart rhythm. In addition, some other functions have been reported for Cx43 that go beyond its role in intercellular communication. Some of these functions have been related to the presence of Cx43 in the inner mitochondrial membrane (mCx43). Strong evidence indicates that mCx43 modulates mitochondrial respiration, ROS production, ATP synthesis, and K+ influx via mitochondrial ATP-dependent K+ channels. The involvement of Cx43 in cardioprotection has been well demonstrated in recent decades and many studies reported that variations in Cx43 expression and localization are part of defence mechanisms put in place by cardiac cells to counteract the propagation of chemotherapeutic drug-induced cardiotoxic effects.

Our results showed that Simvastatin co-treatment significantly reduced Doxorubicin-induced mCx43 over-expression and significantly increased the membrane levels of Cx43 phosphorylated on Ser368. We hypothesized that the reduced expression of mitochondrial Cx43 could justify the reduced levels of calcium stored in mitochondria and the consequent induction of apoptosis observed in the Simvastatin co-treated cells. Moreover, the increased membrane levels of Cx43 phosphorylated on Ser368, that is responsible for the closed conformational state of Gap Junction, let us to hypothesized that Simvastatin leads to cell-to-cell communication interruption to block the propagation of Doxorubicin-induced harmful stimuli.

In conclusion, the novelty of our paper is that we have focused our attention on the ability of Simvastatin to strengthen the cardio-protection mechanisms that are in place in the early stages of damage induced by Doxorubicin (in our experimental model, in fact, treatment with chemotherapeutic drug is 20 hours)

Reviewer 3 Report

I see the authors did great improvements in the manuscript and its now suitable for publication in IJMS

Author Response

Thank you for your previous suggestions which have allowed us to improve our work